UK biobank; COVID-19; SARS-CoV-2; precision medicine

**Corresponding author:**
Qi Feng;
Email: Qi.Feng@ndph.ox.ac.uk

**Topics:** data science **Subtopics:** multi-omics, imaging, big data.

# UK biobank: Enhanced assessment of the epidemiology and long-term impact of coronavirus disease-2019

Qi Feng[1,2] 🟢, Ben Lacey[1,2], Jelena Bešević[1,2], Wemimo Omiyale[1,2], Megan Conroy[1,2], Fenella Starkey[1,2], Catherine Calvin[1,2], Howard Callen[1,2], Laura Bramley[1,2], Samantha Welsh[2], Allen Young[1,2], Mark Effingham[2], Alan Young[1,2], Rory Collins[1,2], Jo Holliday[1,2] and Naomi Allen[1,2]

[1]Oxford Population Health, Clinical Trial Service Unit and Epidemiological Studies Unit (CTSU), Nuffield Department of Population Health, University of Oxford, Oxford, UK and [2]UK Biobank, Stockport, Greater Manchester, UK

## Abstract

UK Biobank is an intensively characterised prospective cohort of 500,000 adults aged 40–69 years when recruited between 2006 and 2010. The study was established to enable researchers worldwide to undertake health-related research in the public interest. The existence of such a large, detailed prospective cohort with a high degree of participant engagement enabled its rapid repurposing for coronavirus disease-2019 (COVID-19) research. In response to the pandemic, the frequency of updates on hospitalisations and deaths among participants was immediately increased, and new data linkages were established to national severe acute respiratory syndrome coronavirus 2 (SARS-CoV-2) testing and primary care health records to facilitate research into the determinants of severe COVID-19. UK Biobank also instigated several sub-studies on COVID-19. In 2020, monthly blood samples were collected from approximately 20,000 individuals to investigate the distribution and determinants of SARS-CoV-2 infection, and to assess the persistence of antibodies following infection with another blood sample collected after 12 months. UK Biobank also performed repeat imaging of approximately 2,000 participants (half of whom had evidence of previous SARS-CoV-2 infection and half did not) to investigate the impact of the virus on changes in measures of internal organ structure and function. In addition, approximately 200,000 UK Biobank participants took part in a self-test SARS-CoV-2 antibody sub-study (between February and November 2021) to collect objective data on previous SARS-CoV-2 infection. These studies are enabling unique research into the genetic, lifestyle and environmental determinants of SARS-CoV-2 infection and severe COVID-19, as well as their long-term health effects. UK Biobank's contribution to the national and international response to the pandemic represents a case study for its broader value, now and in the future, to precision medicine research.

## Impact statement

The existence of a prospective cohort study as large and detailed as the UK Biobank, with high degrees of participant engagement, enabled its rapid repurposing in 2020 to support coronavirus disease-2019 (COVID-19) research. In response to the pandemic, UK Biobank increased the frequency of health outcome updates and established new data linkages with national severe acute respiratory syndrome coronavirus 2 (SARS-CoV-2) testing and primary care records. These linkages were supplemented by several sub-studies to enable innovative COVID-19 research. For example, UK Biobank is currently the only resource that provides large-scale data to investigate the impact of SARS-CoV-2 infection on multi-organ pathophysiology based on standardised structural and functional imaging scans before and after infection. These enhancements, combined with the data collected by the study before the pandemic on genomics, metabolomics, lifestyle, and the environment, make UK Biobank uniquely placed to allow scientists worldwide to answer questions about the determinants and wide-ranging health consequences of SARS-CoV-2 infection. Importantly, UK Biobank resources are made available to researchers around the world, allowing a wide range of research to be conducted in the public interest. UK Biobank's contribution to the response to the pandemic presents a case study for the broader value of the resource, and other blood-based prospective cohort studies ('biobanks'), for precision medicine research.

## Introduction

Prospective cohort studies with long-term storage of biological specimens ('biobanks') have proved particularly valuable for investigating the causes of disease and advancing precision

medicine research (Denny and Collins, 2021). UK Biobank is an intensively characterised prospective cohort study of 500,000 adults (Sudlow et al., 2015). It was established by the Medical Research Council and Wellcome Trust (with additional support from other funders) to enable approved academic and commercial researchers worldwide to conduct health research in the public interest.

UK Biobank's large size, depth of participant characterisation, extent of follow-up, and ease of access by researchers, has made it a key biomedical resource for public health research globally. By the end of 2022, there were over 30,000 incident cases of diabetes, 25,000 cases of depression, 15,000 cases of myocardial infarction and 10,000 cases of breast cancer, highlighting the growing value of the resource to a wide range of diseases. The prospective study design enables risk factors to be assessed before the disease develops, which avoids major bias from reverse causation. In addition, the breadth of data provided by participants allows researchers to investigate the relevance of many different types of genetic, physiological, lifestyle and environmental exposures for the development of life-threatening and disabling diseases of middle and old age.

The existence of such a large and detailed cohort, with a high degree of participant engagement, enabled its rapid repurposing at the start of the pandemic to support COVID-19 research. This review describes UK Biobank's contribution to the national and international response to the pandemic, and in doing so, highlights the study's broader value, now and in the future, to understanding the causes and consequences of major diseases, and to the advancement of precision medicine research. The review first summarises the participant recruitment and data collection in UK Biobank, and provides an update on enhancements to the study prior to the pandemic. It then describes UK Biobank's efforts to enhance the resource specifically for COVID-19 research, together with some initial research findings to illustrate the value of this data.

## UK biobank

### Recruitment of participants

Potentially eligible participants were identified through National Health Service (NHS) central registries and invited to attend one of the 22 UK Biobank assessment centres located within about 30 miles of their home address. The assessment centres were located throughout England, Scotland and Wales, in both rural and urban areas, including those with a high proportion of ethnic minority populations. A total of 502,000 participants aged 40–69 years were recruited between 2006 and 2010. The age range for inclusion represented a pragmatic compromise between participants being young enough for the initial assessment to take place before the disease was likely to have had a material impact on exposures, and old enough for sufficient incident health outcomes to occur in the first few decades of follow-up.

### Baseline assessment

The baseline assessment comprised a self-administered touch-screen questionnaire, a brief interview, physical measurements, and the collection of blood, urine and – in a subset – saliva samples for long-term storage (Figure 1, Table S1 in the Supplementary Material). The touch-screen questionnaire included questions on a wide range of exposures, including sociodemographic factors, lifestyle, health status, family history and environmental exposures. This was followed by a computer-assisted interview administered by a nurse to obtain more detailed information on medical history.

Physical measurements taken included blood pressure, heart rate, spirometry, grip strength and anthropometry. A large subset of participants also underwent an eye examination and tests for hearing, cardiorespiratory fitness, calcaneal bone density and arterial stiffness. All participants provided consent for the use of their de-identified data for health-related research and for UK Biobank to access their medical and other health-related records, as well as permission to re-contact them for further data collection.

### Follow-up for health outcomes

Participants are followed up for health outcomes through linkage to national death and cancer registries, hospital inpatient admissions and primary care records (available up until 2017 for approximately 45% of the cohort) (Table S2 in the Supplementary Material). In addition, UK Biobank periodically invites participants to complete web-based questionnaires to obtain information on health-related issues that are not captured well in linked medical records (such as cognitive function, pain, mental health and well-being) (Table S1 in the Supplementary Material).

### Study enhancements

Since recruitment, UK Biobank has continued to collect new data on its participants. Between 2012 and 2013, 20,000 participants attended a repeat assessment visit (including repeat sample collection), principally to enable researchers to identify and correct for regression dilution bias (caused by measurement error and within-person variation in exposure levels) in their analyses (Clarke et al., 1999). Objectively measured physical activity data were also collected from 100,000 UK Biobank participants during 2013 and 2016 using a wrist-worn accelerometer worn continuously for 7 days, with repeat assessments conducted in a subset (2,500 participants) in 2018. Web-based questionnaires have also been used to collect more detailed information on particular exposures of interest, such as diet, pain, mental health and occupational history.

Since 2014, UK Biobank has been conducting a multimodal imaging study in up to 100,000 of its participants (Littlejohns et al., 2020). This includes a repeat of the baseline assessment together with magnetic resonance imaging (MRI) scans of the heart, abdomen and brain, a whole-body dual-energy X-ray absorptiometry (DXA) scan, a carotid ultrasound scan, a 12-lead ECG and, in a subset of older participants, continuous cardiac monitoring for 14 days. Up to 60,000 of these participants will also be included in a second imaging assessment over the coming years to enable the relevance of changes in measures of internal organ structure and function to be assessed.

UK Biobank's policy is to conduct cohort-wide measurement of biomarkers, wherever possible. In contrast to generating biomarker data that are relevant to one particular health outcome (such as is done in a nested case–control approaches), cohort-wide measures can support a very wide range of research on many different diseases by many different researchers. They also facilitate precision medicine approaches by enabling researchers to examine the consistency of associations in subgroups of the population (e.g., by age, sex, socioeconomic status, etc.) or by levels of other risk factors (Allen et al., 2020). Cohort-wide data are available in UK Biobank for key haematological and biochemical markers, leukocyte telomere length, genome-wide genotyping, whole exome sequencing and, in late-2023, whole genome sequencing. Nuclear magnetic resonance (NMR) metabolomics data are currently available for 120,000 participants, with cohort-wide data expected to be made

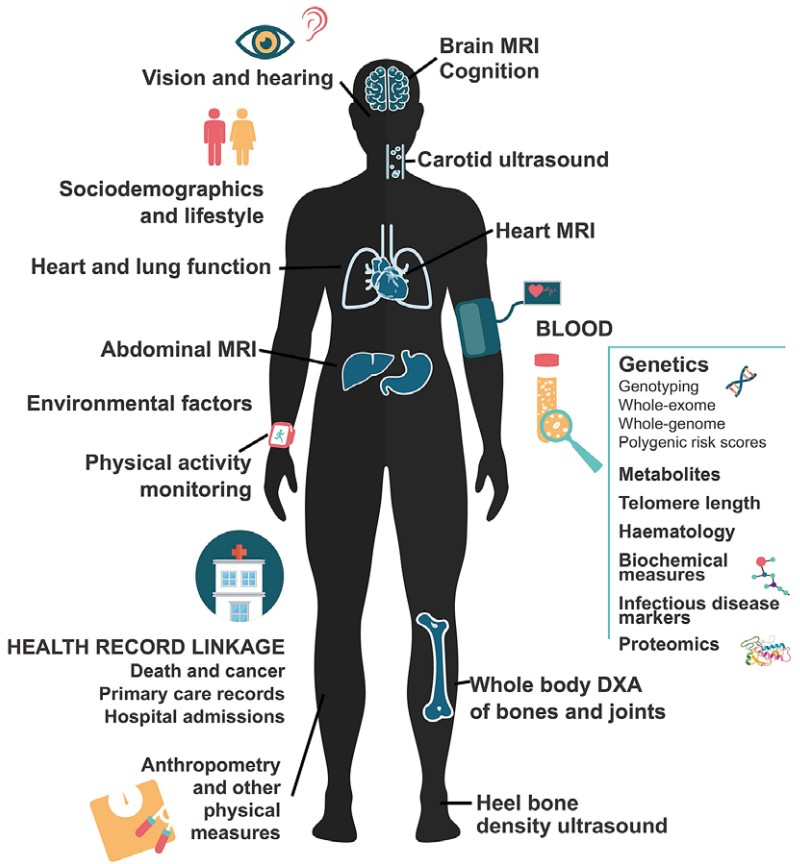

**Figure 1.** UK Biobank data.
At recruitment, participants completed questionnaires on a wide range of exposure; physical measurements were taken including blood pressure, heart rate, spirometry, grip strength and anthropometry; and blood, urine and – in a subset – saliva samples were collected for long-term storage. A multimodal imaging sub-study in up to 100,000 participants includes a magnetic resonance imaging (MRI) scan of the heart, abdomen and brain, whole-body dual-energy X-ray absorptiometry (DXA) scan, carotid ultrasound and 12-lead ECG. Participants are followed up for health outcomes through linkage to national death and cancer registries, hospital inpatient admissions and (for a subset) primary care records.

available in due course. Proteomic measurements using the 3,000 protein O-link platform for approximately 60,000 participants will be made available by late-2023, with the possibility of proteomic measurements being extended to the full cohort at a later stage. (Table 1).

### Access to the resource

UK Biobank data are available to all bona fide researchers world-wide to perform health-related research, irrespective of whether they are based at an academic, charitable or commercial institution, with no preferential or exclusive access (Conroy et al., 2019). To access the resource, researchers must first register with UK Biobank and then submit an application outlining their research and how it would benefit public health. UK Biobank also encourages applications for the analysis of biological samples and other proposals for enhancing the characterisation of participants or their health outcomes. Students are able to access the data at a substantially reduced cost and industry grants are available to cover the access costs for applicants from low- and middle-income countries. Researchers are also able to access and analyse the UK Biobank data via the

Research Analysis Platform (RAP), an online informatics platform that allows approved researchers to access and analyse the entire UK Biobank database securely, in the cloud, from anywhere in the world. To further democratise access to this resource, research credits have been made available for early-career researchers and those from low- and middle-income countries. UK Biobank now has over 30,000 registered researchers (75% from outside of the UK) and 3,000 approved projects. Since 2012 (when data were first made available for research), the UK Biobank community of researchers has published more than 6,000 research articles, with over 178,000 citations. UK Biobank has also been referenced in over 500 patent applications.

### Study enhancements to enable COVID-19 research

#### Enabling research into the determinants and complications of severe COVID-19

UK Biobank made efforts very early in the SARS-CoV-2 pandemic to support research into the determinants of severe COVID-19. At the start of the pandemic in March 2020, the frequency of health

**Table 1.** Biological sample assay data available in UK Biobank

| Data type | Details | Number of participants | Date of collection |
|---|---|---|---|
| Biochemistry markers | 34 biomarkers were assayed in plasma, serum, red blood cells, and urine samples; includes established risk factors for disease (e.g., lipids for vascular disease, sex hormones for cancer), diagnostic measures (e.g., HbA1c for diabetes and rheumatoid factor for arthritis), and other measures (such as liver and renal function tests) | Whole cohort (baseline) | 2006–2010 |
| | | 20,000 (first repeat assessment) | 2012–2013 |
| Infectious agents | Antibody sero-positivity status of 20 pathogens was measured | 10,000 (baseline) | 2006–2010 |
| | | 537 (first repeat assessment) | |
| Genotyping | Genome-wide genotyping was performed using the UK BiLEVE Axiom array (approximately 50,000 participants) and the UK Biobank Axiom Array (approximately 450,000 participants). Approximately 850,000 variants were directly measured, with >90 million variants imputed | Whole cohort (baseline) | 2006–2010 |
| Whole exome sequencing | Whole-exome sequencing measures the regions of the genome (about 2%) that are involved in coding for proteins, and is particularly suitable for identifying disease-causing and/or rare genetic variants | Whole cohort (baseline) | 2006–2010 |
| Whole genome sequencing | Whole genome sequencing measures the entire genome and will provide information that will complement and enhance the existing genotyping and exome data | 200,000 (baseline)* | 2006–2010 |
| Telomeres | Chromosomal telomere length | Whole cohort (baseline) | 2006–2010 |
| | | 20,000 (first repeat assessment) | 2012–2013 |
| Plasma metabolites | >200 circulating metabolites (predominantly lipids) were measured using NMR metabolomics platform | Whole cohort (baseline) | 2006–2010 |
| | | 20,000 (first repeat assessment) | 2012–2013 |
| Plasma proteins | Approx. 3,000 circulating proteins were measured | 60,000 (baseline)† | 2006–2010 |

Abbreviations: HbA1c, haemoglobin A1c test; NMR nuclear magnetic resonance. *Data on the whole cohort expected to be available in late 2023. †Data on 1,500 proteins available in Spring 2023 and data on 3,000 proteins expected to be available mid-2023.

outcome updates on hospitalisations and deaths was immediately increased, and new data linkages were established to SARS-CoV-2 testing data for the full cohort. Owing to the high interest in understanding the determinants of severe COVID-19, emergency legislation was introduced (via a Control of Patient Information (COPI) notice issued by the Secretary of State for Health and Social Care) that allowed UK Biobank to access primary care data for all participants resident in England (about 80% of the cohort) for the purpose of COVID-19 research (Table S2 in the Supplementary Material). To ensure research could be performed as quickly as possible, expedited access was provided to the established community of approved investigators to use the resource for COVID-19 research, facilitating a wide variety of research studies. By the end of 2022 there were about 250 COVID-19 peer-reviewed research publications based on UK Biobank data, with more than 4,500 citations (Figure 2, Table S3 in the Supplementary Material). This has included the investigation of the associations of socio-demographic (e.g., age, sex, ethnicity, socioeconomic status, household size), lifestyle (e.g., shift-work, smoking, alcohol consumption, physical activity), environmental (e.g., air pollution) and clinical factors (as assessed from health linkage data and imaging scans), as well as biomarkers (e.g., telomere length and circulating metabolites) with the risk of severe COVID-19 (Ho et al., 2020; Kolin et al., 2020; McQueenie et al., 2020; Raisi-Estabragh et al., 2020; Atkins et al., 2021; Fatima et al., 2021; Julkunen et al., 2021; Peters et al., 2021; Travaglio et al., 2021; Wang et al., 2021; Gillies et al., 2022; Sheridan et al., 2022). This work has highlighted the particular importance of age, ethnicity, smoking, obesity and major

co-morbidities (including cardiovascular and renal disease) as risk factors for severe outcomes following SARS-CoV-2 infection.

The depth of characterisation of participants in the UK Biobank allows researchers to explore in detail the potential biological pathways between risk factors and severe COVID-19. In particular, the genetic data available for all 500,000 participants have enabled the assessment of the causality of such risk factors using Mendelian randomisation (MR) approaches. MR takes advantage of the random assortment of genes from parents during gamete formation and conception to mimic the effect of a randomised controlled trial for a particular exposure in observational studies (Smith and Ebrahim, 2003). Such analyses support a causal association of several major modifiable risk factors with severe COVID-19, including obesity and smoking (Li and Hua, 2021; Clift et al., 2022). For example, a recent study of both observational and MR associations of body composition, fat distribution and metabolic consequences of excess adiposity with severe COVID-19 outcomes found robust associations with general adiposity (including body mass index) but not with central adiposity or some of the metabolic consequences of excess adiposity, such as diabetes (Gao et al., 2022).

Genetic data in UK Biobank have also been used to conduct genome-wide association studies (GWAS) to identify novel genetic variants (single nucleotide polymorphisms (SNPs)) associated with an increased risk of severe COVID-19. A GWAS analysis on UK Biobank published early during the pandemic identified eight genetic variants that significantly increased the risk of COVID-19 mortality (Hu et al., 2021). These variants have been associated with pulmonary cilia dysfunction, cardiovascular disease, thromboembolic

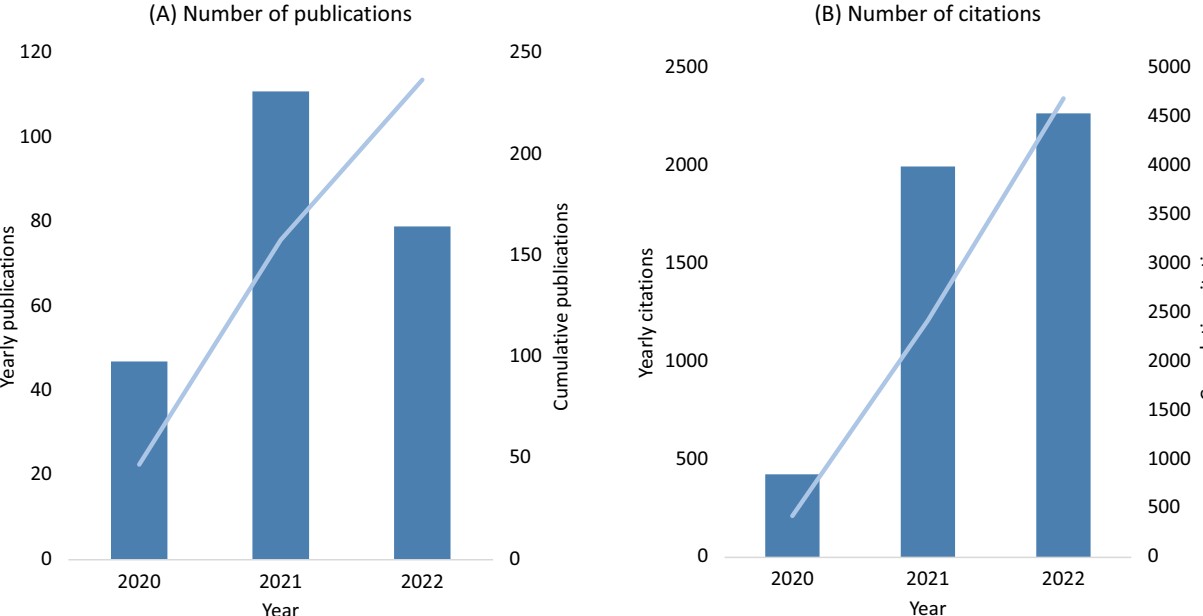

**Figure 2.** UK Biobank publications on COVID-19 (A) and related citations (B).

disease, mitochondrial dysfunction and the innate immune system dysfunction (Hu et al., 2021). This analysis, together with other large meta-analyses of GWAS projects using UK Biobank data (Initiative, 2021; Pairo-Castineira et al., 2021) provided timely insights into the pathogenesis of severe COVID-19 (Thibord et al., 2022).

Researchers have also used the SNPs identified in GWAS to construct polygenic risk scores for severe COVID-19. Polygenic risk scores combine the effect of a set of genetic variants – each of which on their own has a small effect on risk – to obtain a measure that has sufficient information to identify those at high genetic risk of developing a certain health outcome (Lewis and Vassos, 2020). For example, a study using UK Biobank data found that adding a polygenic risk score to a reference prediction model of clinical risk factors (age, ethnicity, Townsend deprivation index, body mass index, smoking and baseline comorbidities) for severe COVID-19 improved the area under the curve value (a measure of the overall performance of the model) from 0.72 to 0.79 (Dite et al., 2021). This study highlights the potential clinical value of genetic data in identifying individuals who are most likely to benefit from targeted intervention following SARS-CoV-2 infection, such as the use of antivirals.

Several studies have used UK Biobank data to evaluate the associations between specific genetic variants and COVID-19 outcomes to identify risk factors or evaluate potential therapeutic targets. For example, the calcium ion channel gene *ORAI1* is known for its role in immune response, inflammation, platelet activation and thrombus formation, and was therefore considered as a potential drug target for COVID-19. However, a UK Biobank study found that genetic variants within *ORAI1* were not associated with severe COVID-19 (Shawer et al., 2022) consistent with observational analyses that found no association between calcium channel blocker use and COVID-19 outcomes (Alsagaff et al., 2021). Other analyses using UK Biobank data have reported that the *ApoE* e4/e4 genotype (which not only affects lipoprotein function but also moderates macrophage inflammatory phenotypes) increases the risk of severe COVID-19, independently of pre-existing dementia, cardiovascular disease and type 2 diabetes (Kuo et al., 2020). Taken

together, this body of work indicates the potentially important biological pathways through which SARS-CoV-2 infection can lead to severe COVID-19, and has highlighted several potential therapeutic targets.

Enhanced cohort-wide linkage of health outcomes and test results enable researchers to assess the longer-term health impacts of SARS-CoV-2 infection across the full disease spectrum (i.e., ranging from those who were asymptomatic to those with severe COVID-19). For example, previous research based on the linked healthcare data in UK Biobank showed that participants with severe COVID-19 (i.e., individuals who were hospitalised with the condition) had an increased risk of a range of cardiovascular outcomes, whereas less severe disease was associated with an increased risk of venous thromboembolism but not with other cardiovascular-specific outcomes (Raisi-Estabragh et al., 2022).

### Enabling research into the distribution and determinants of SARS-CoV-2 infection

At the request of the UK Government and with funding from the Department of Health and Social Care, UK Biobank established the in 2020 (Figure S1 in the Supplementary Material). The study aimed to use the UK Biobank cohort to improve understanding of the distribution and determinants of SARS-CoV-2 infection, and to assess the long-term persistence of SARS-CoV-2 antibodies following infection (https://www.gov.uk/government/publications/uk-biobank-covid-19-antibody-study-final-results/uk-biobank-covid-19-antibody-study-final-results. Accessed 02/07/2023).

From May to November 2020, monthly blood samples were collected from 11,000 UK Biobank participants and 9,000 of their adult children and grandchildren, ensuring representation across different geographical locations, age groups, sex, region and socio-economic status. Participants were asked to provide a finger-prick capillary blood sample at home, at approximately monthly intervals on six occasions, and to complete a questionnaire about potential symptoms of COVID-19. Participants returned the blood samples to UK Biobank, and these were then processed and transported to

the Target Discovery Institute (University of Oxford) for measurement of IgG antibodies to the spike protein (IgG-S) of SARS-CoV-2; 18,887 individuals (94%) provided at least one sample that was successfully assayed.

In January 2021, the participants were asked to complete a further questionnaire on social and environmental factors that may have affected their risk of exposure to SARS-CoV-2, such as employment, lifestyle and household composition, across different periods of the pandemic. A final blood sample was then collected between November 2021 and March 2022 (i.e., approximately 18 months after the first blood sample collection) in order to assess the long-term persistence of SARS-CoV-2 antibodies following infection. As a result of the vaccine rollout at the end of 2020, it was not possible to use IgG-S to determine seroprevalence of SARS-CoV-2 in that final sample, as these antibodies are also generated by the vaccine. Instead, participants were sent a capillary blood sampling kit (the Thriva antibody test) that was used to measure the presence of antibodies to the nucleocapsid protein (IgG-N), which is only produced following infection, but not vaccination.

Seroprevalence estimates by various characteristics (including age, sex, socioeconomic status, ethnicity and UK region) were reported to the UK Department of Health and Social Care on a regular basis during 2020 (https://www.gov.uk/government/publications/uk-biobank-covid-19-antibody-study-final-results/uk-biobank-covid-19-antibody-study-final-results. Accessed 02/07/2023). The percentage of the population with IgG-S antibodies to SARS-CoV-2, indicating past infection, rose from 6.6% to 8.8% between May and November 2020. At the end of the study period, there was no evidence that seroprevalence differed by sex, but it was found to be higher in participants who were younger, of lower socioeconomic status, from urban areas, and from Black and other minority ethnic groups. A key finding was that 88% of participants who had tested positive for previous infection retained antibodies for at least 6 months following infection, suggesting some degree of immunological protection for a period of time. The data generated as part of this study are now integrated into the UK Biobank resource to allow researchers to investigate further the determinants of SARS-CoV-2 antibody response.

This sub-study, together with complementary work on the whole cohort, has helped to clarify some of the major social and environmental determinants of SARS-CoV-2 infection (Niedzwiedz et al., 2020; Yanik et al., 2022). For example, recent analyses have identified a set of major social factors, including occupation and region of residence that account for most ethnic disparities in SARS-CoV-2 infection during the first wave of the pandemic in the UK, highlighting the particular relevance of occupational factors and region of residence in explaining these inequalities (Omiyale et al., 2023).

### Enabling research into longer-term health effects of SARS-CoV-2 infection

To enable comprehensive and objective research into the possible longer-term health effects of SARS-CoV-2, UK Biobank set up the coronavirus self-test antibody study and the COVID-19 repeat imaging study, which will be of value to scientists investigating the longer-term health effects across the full spectrum of COVID-19 disease severity and COVID-19's effect on internal physiology over time (Table 2).

#### Coronavirus self-test antibody sub-study

Between February and July 2021, approximately 450,000 UK Biobank participants living in mainland UK were invited to take part in the coronavirus self-test antibody sub-study (Figure S2 in the Supplementary Material). The study aimed to collect objective evidence of previous SARS-CoV-2 infection from as many UK Biobank participants as possible using a SARS-CoV-2 antibody self-testing kit. Overall, approximately 200,000 UK Biobank participants took part.

Consented participants were sent a self-testing kit to identify the presence of IgG-S antibodies, with the first approximately 50,000 receiving the Fortress Fast COVID-19 device and the remainder the AbC-19™ Rapid Test. Participants were asked to report their antibody test results (IgG-S positive, negative, or invalid) and COVID-19 vaccination status to UK Biobank via an online questionnaire. In order to help minimise false positives with the self-testing kit, participants with a positive result who reported not having had a COVID-19 vaccination were sent a second test kit to confirm the result. Likewise, in order to avoid false positives due to IgG-S antibodies produced by a vaccine, participants who had a positive result following vaccination (approximately 74,000) were consented to receive a capillary blood sampling kit (the Thriva antibody test) and asked to return their sample to UK Biobank by post. Approximately 60,000 samples were received and tested for IgG-N antibodies to identify those with evidence of a previous infection (as opposed to vaccination). It was estimated that approximately 20% of participants were infected among the

**Table 2.** Key enhancements to enable COVID-19 research in UK Biobank

| Enhancement | Details |
| --- | --- |
| Enhanced data linkage | UK Biobank established a linkage to national SARS-CoV-2 testing data. In addition, the study was able to access primary care data for all participants resident in England (~80% of the cohort) under emergency legislation via a Control of Patient Information (COPI) notice issued by the Secretary of State for Health and Social Care; this legislation expired in June 2022. |
| UK Biobank SARS-CoV-2 serology sub-study | This sub-study collected blood samples from approximately 11,000 UK Biobank participants and approximately 9,000 of their adult children and grandchildren. It aimed to improve understanding of the distribution and determinants of SARS-CoV-2 infection and assess the persistence of SARS-CoV-2 antibodies following infection. |
| UK Biobank coronavirus self-test antibody sub-study | This sub-study aimed to collect objective evidence of previous SARS-CoV-2 infection from as many UK Biobank participants as possible using a SARS-CoV-2 antibody self-testing kit. In total, approximately 200,000 UK Biobank participants took part in the study. |
| UK Biobank COVID-19 repeat imaging sub-study | This study aimed to facilitate research into the short- to medium-term effects of SARS-CoV-2 infection on the structure and function of major organs. The sub-study recruited approximately 2,000 participants in 2021 who had been imaged prior to the pandemic – half of whom had evidence of SARS-CoV-2 infection, and half of whom did not. |

200,000 participants. Data from this coronavirus self-test antibody sub-study were made available in February 2022.

### COVID-19 repeat imaging sub-study

Before the COVID-19 pandemic, UK Biobank was halfway through performing multimodal imaging of up to 100,000 participants. This enabled UK Biobank to undertake a unique repeat imaging sub-study, with the aim of building a resource to enable research into the extent to which SARS-CoV-2 infection is associated with changes in the structure and function of major organs over the short to medium term. The sub-study recruited approximately 2,000 participants in 2021 who had been imaged prior to the pandemic, half of whom had evidence of SARS-CoV-2 infection ('cases'), and half of whom did not ('controls') (Douaud et al., 2022).

Participants were eligible for the COVID-19 repeat imaging study if high-quality scans had been obtained from their first imaging assessment and there had been no incidental findings. Cases were defined as individuals with a previous SARS-CoV-2 infection, identified via linkage to medical records or a positive antibody test (as assessed in the coronavirus self-test antibody study). Controls were eligible individuals with no evidence of previous SARS-CoV-2 infection, and were matched to each case based on sex, ethnicity (White/Non-White), date of birth (+/−6 months), location of imaging assessment clinic and date of first imaging assessment (+/− 6 months). The mean age of participants at their repeat imaging assessment was 62 years and the average duration between imaging assessments was 3 years.

The imaging data on all approximately 2,000 participants are now available to the research community. Emerging results from the brain images taken before and after SARS-CoV-2 infection have found evidence of brain-related abnormalities, including a greater reduction in grey matter thickness and tissue contrast in the orbitofrontal cortex and parahippocampal gyrus (which have roles in memory and cognition) (Douaud et al., 2022). There were also changes indicative of tissue damage in regions that are functionally connected to the primary olfactory cortex (concerned with the sense of smell). However, these differences were, on average, modest and it remains unclear whether these effects persist over the long term. In contrast, research using the cardiac images from the COVID-19 repeat imaging study has not found clinically significant persistent cardiac pathology in the UK Biobank population after generally milder (non-hospitalised) SARS-CoV-2 infection (Bai et al., 2021). Other imaging studies investigating the effect of SARS-CoV-2 on internal physiology have only collected scans post-infection (and are largely focused on those with severe disease), so they cannot assess whether the infection has caused a direct effect on internal organs.

### Opportunities and limitations of UK biobank

The key strengths of UK Biobank include its large sample size, long follow-up period, as well as the breadth and depth of data collected, which make it, at present, a uniquely important biomedical resource to investigate the determinants of major diseases. The value of the resource will continuing to grow over the coming years as the number of incident disease event accrue, and the study is further developed, such as through the release of whole-genome sequencing data on the whole cohort later in 2023. The enhancements to the study in response to the SARS-CoV-2 pandemic (including the sub-studies described above) have generated novel insights into the causes and consequences of COVID-19, but, importantly, this work also serves to illustrate the potential of the study to rapidly advance understanding of a particular disease with focused investment and researcher interest.

There are, however, some key limitations to the study. First, although UK Biobank has updated linkage to death registries, cancer registries, and hospital inpatient admission data for all participants, linkage to primary care data is currently only for 45% of the cohort available up until 2017; the emergency access to primary care data for all participants resident in England for COVID-19 related research expired in July 2022. This restricted access of primary care data limits case ascertainment as well as the range of diseases that can be studied. Second, UK Biobank is not a nationally representative cohort (Fry et al., 2017). UK Biobank participants are, on average, less deprived, and more likely to be of White ethnicity and somewhat healthier, than the general UK population. As such, UK Biobank estimates of incidence or prevalence should not be generalised to the wider population, but disease association estimates are likely to be generalizable, and the study is sufficiently large that heterogeneity by population subgroup can usually be assessed. Third, UK Biobank does not have sufficient cases of some diseases for reliable analyses. This includes diseases that are rare in the UK population or those that occur largely outside the age range of participants in the study.

### Conclusion

The existence of a prospective cohort as large and detailed as the UK Biobank, with a high degree of participant engagement, enabled its rapid repurposing in 2020 to support COVID-19 research. In response to the pandemic, UK Biobank increased the frequency of health outcome updates, and established new data linkages with national SARS-CoV-2 testing and primary care records. These linkages were supplemented by several sub-studies to enable novel COVID-19 research (described comprehensively for the first time in this paper). Currently, UK Biobank is the only resource that provides large-scale data to investigate the impact of SARS-CoV-2 infection on multi-organ pathophysiology based on standardised structural and functional imaging scans before and after infection. The data from these enhancements, combined with the data on genomics, metabolomics, lifestyle, and the environment collect by the study, make UK Biobank uniquely placed to allow scientists worldwide to answer questions about the determinants and wide-ranging health consequences of SARS-CoV-2 infection. Importantly, UK Biobank resources are made available to researchers around the world, allowing a wide range of different types of research to drive improvements in population health. UK Biobank's contribution to the national and international response to the pandemic represents a case study for the broader value and potential of the resource for precision medicine research.

**Open peer review.** To view the open peer review materials for this article, please visit http://doi.org/10.1017/pcm.2023.18.

**Supplementary material.** The supplementary material for this article can be found at http://doi.org/10.1017/pcm.2023.18.

**Data availability statement.** UK Biobank is available for open access, without the need for collaboration, to any bona fide researcher who wishes to use it to conduct health-related research for the benefit of the public.

**Acknowledgments.** We would like to thank the UK Biobank study participants.

**Author contribution.** Q.F., B.L., J.B., W.O., and M.C. wrote the first draft of the manuscript. All authors provided critical comments and suggestions. All authors have agreed on this submission.
Q.F., B.L., J.H., N.A. equal contribution to this article.

**Financial support.** The funding for the UK Biobank SARS-CoV-2 serology study was provided by the United Kingdom Department of Health and Social Care. The core funding for UK Biobank is provided by the UK Medical Research Council, Wellcome, British Heart Foundation, Cancer Research UK, and National Institute for Health Research (grant ref. 223,600/Z/21/Z). The Clinical Trial Service Unit and Epidemiological Studies Unit (CTSU), which is part of the Nuffield Department of Population Health, receives research grants from industry that are governed by University of Oxford contracts that protect its independence and has a staff policy of not taking personal payments from industry; further details can be found at https://www.ndph.ox.ac.uk/files/about/ndph-independence-of-research-policy-jun-20.pdf. The funders had no role in the design of the study; in the collection, analysis or interpretation of data; in the writing of the report; or in the decision to submit the paper for publication.

**Competing interest.** The authors have no disclosures relevant to this study.

**Ethics standard.** A favourable ethical opinion was provided by the North West (Haydock) Research Ethics Committee (ref: 16/NW/0274). All participants provided informed consent.

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
