## [Reviewer Report]

The UK Biobank is a very important resource for science and public health and has already proven its worth. It has also been shown to be an essential source of information in the COVID-19 pandemic.

General Comment: The problem with the current version is that it reads like a sales pitch, without explaining the rationale for this article and weighing opportunities, successes, and limitations.

Introduction. It is not clear to me what the purpose of the paper is. I understand that the manuscript is about the description of the UK Biobank and its value for COVID-19 research, but please explain the purpose of the manuscript in the introduction.

UK Biobank section. There are other publications that describe the UK Biobank in detail. What does this manuscript add to these existing publications?

The Impact and Introduction section seems to focus on the value of UK Biobank for COVID-19 research. This section describing the UK Biobank is more general and does not mention COVID-19 at all. Please make clear to the reader that this section is only about the pre-existing data.

The “Research Enhancements to Enable COVID-19 Research” section describes the data collections and data linkages for COVID-19 research, but it is mixed with results. For example: “By the end of 2022 there were approximately 250 COVID-19 peer-reviewed research publications based on UK Biobank data, with over 4500 citations”. A separation of the the data collection and the results would give the manuscript more structure and a more objective, academic perspective. What data collections or linkages failed or were not possible? What was not possible with the data? What were limitations? It is very important for future users of the UK Biobank to understand its limitations. This does not make the UK Biobank less valuable, but does contribute to its reliability.

---

## [Editor Report]

The topic and content of this review is very good and interesting, and the topic is one that we really want to publish, however the first reviewer has a solid point and I think there needs to be a revision. Simply, there seems to me too much un-cited (presumably unpublished) results in this manuscript to be a review.

Given the time that has passed, hopefully more has been published to make a revision easier with the simple addition of citations (eg, the Olink study mentioned). Otherwise, I think most/all of the remaining uncited results could fit the scope of a review if the authors were to show a bit more of the workings, and then tie it back to something published. (perhaps a table of the 250 published works based on the program, as a supplementary?) I believe that’s the other reviewer’s general point too, to make it more clear what is results versus review; and then to balance out the whole review a bit.

---

## [Editor Report]

An important contribution to precision medicine, combining the power of a large-scale population study in the context of the SARS-CoV2 pandemic.